# Effects of a social support family caregiver training program on changing blood pressure and lipid levels among elderly at risk of hypertension in a northern Thai community

Sorawit Boonyathee[1¤], Katekaew Seangpraw[2¤]*, Parichat Ong-Artborirak[3], Nisarat Auttama[2¤], Prakasit Tonchoy[2¤], Supakan Kantow[2¤], Sasivimol Bootsikeaw[2¤], Monchanok Choowanthanapakorn[2¤], Pitakpong Panta[4¤], Dech Dokpuang[5¤]

1 School of Medicine, University of Phayao, Phayao, Thailand, 2 School of Public Health, University of Phayao, Phayao, Thailand, 3 Faculty of Public Health, Chiang Mai University, Chiang Mai, Thailand, 4 School of Nursing, University of Phayao, Phayao, Thailand, 5 School of Allied Health Sciences, University of Phayao, Phayao, Thailand

¤ Current address: University of Phayao, Phayao, Thailand
* eungkaew@gmail.com

## Abstract

Hypertension is becoming increasingly prevalent among the elderly. Family caregivers play an important role in caring for elderly people and empowering them to care for themselves. This study's goal was to see how social support training for family caregivers affected changes in hypertension, total cholesterol, and high-density lipoprotein (HDL), and how such support led to the prevention of hypertension behaviors among the elderly in rural areas. This was a quasi-experimental study with 268 elderly people at risk of hypertension and their caregivers. Sixty seven pairs of elderly people and their caregivers were assigned to the intervention and control groups. Baseline data were collected in November 2020. The intervention group received the Social Support Family Caregiver Training Program (SSFCTP), while the control group received a regular program from the local health authority. The activity lasted 12 weeks, with home visits and telephone check-ups along the way, and data collection took place after the program ended. The final data were collected three months after the end of the intervention. An analysis of repeated measures ANOVA showed the overall effect of the SSFCTP on knowledge, self-efficacy, health care behaviors, and blood pressure among elderly people during three different time periods (p<0.05). Furthermore, the intervention program had a time-dependent effect on knowledge, blood pressure, and total cholesterol levels (p<0.05). In terms of caregiver outcomes, there was an overall difference among the degrees of knowledge, self-efficacy, and behaviors toward health care displayed by elderly hypertensive patients during the three different time periods (p<0.05). The average knowledge and self-efficacy of the participants improved after the intervention. As a result, better self-care behaviors and lower blood pressure and total cholesterol levels were observed among the elderly participants after the intervention. The programs emphasized the importance of caregivers' roles in providing social support, boosting confidence,

**Data Availability Statement:** All relevant data are within the paper and its Supporting Information files (S3 File).

**Funding:** The authors are grateful to the research project was supported by the Thailand science research and innovation fund and the University of Phayao the Unit of Excellence named "Health Promotion and Quality of Life" grant number FF64-UoE009. The funders had no role in study design, data collection and analysis, decision to publish, or preparation of the manuscript.

**Competing interests:** The authors have declared that no competing interests exist.

and encouraging participation in caring, monitoring, and assisting the elderly in controlling blood pressure and other health issues.

## Introduction

High blood pressure is one of the most common risk factors for cardiovascular diseases, including ischemic heart disease and stroke, which account for 9.4 million deaths worldwide every year [1, 2]. According to the World Health Organization, hypertension affects the health of 22 percent of the adult population worldwide [2]. Studies show that the number of people with hypertension continues to increase; and it is also one of the most common chronic health problems when it comes to health service and consultation with a doctor [3, 4]. Hypertension is prevalent in the Thai population among those aged 15–79 -years old, and statistics show it increased from 15.3% in 2015 to 16.5% in 2018. By 2019, the prevalence had risen among people aged 55 to 64, 65 to 74, and 75 and up (28.7%, 39.0%, and 44.1%, respectively) [5]. The increase in hypertensive patients in Thailand is often seen among the elderly, particularly in Phayao province [6].

The province is located in northern Thailand, a largely mountainous area. The community is rural, and retains the traditions and culture of the Lanna community [6]. In 2017, the prevalence of hypertension in this area increased by 19.5% and increased a further 36.2% in 2018 [6, 7]. In addition, non-communicable disease (NCD) data for hypertensive patients aged 60–70 years with complications show that older adults are at higher risk of developing diseases such as cardiovascular disease and coronary disease (26.5%) [7, 8]. Based on provincial health-screening data for NCDs among patients, it was discovered that the health results did not meet the standard criteria, owing to lifestyle changes, such as lack of physical activity, the consumption of high- sodium, and high-fat diets, drinking alcohol, and smoking, all of which contributed to an increase in hypertension and health-related diseases [7]. Hypertension among older Thai adults may create complications such as diabetes, stroke, and kidney failure [5]. Complications of the disease can cause long-term health issues and affect one's ability to perform daily activities such as eating, bathing, moving, and bowel movements, as well as social activities in which the elderly should engage when they are not at home [9]. As a result of such illnesses, a person's reliance and care is placed on family members, who are the elderly's closest source of social support.

The family institution is the primary source of social support in caring for the elderly's physical, emotional, and social needs [9, 10]. From literature reviews, most elderly caregivers are family members such as a spouses and offspring [10]. Therefore, older adults with hypertension need healthcare from their caregivers, particularly family, community, and volunteer caregivers [9, 10]. Several studies have examined the factors that influence hypertension treatment and prevention behaviors, and have found that hypertensive patients' behaviors are directly linked to family support [11, 12]. The evidence indicates that family advocacy is a vital strategy to help drive family members to think, make decisions, monitor, and solve family problems [13]. We found that social support for, and participation among, the elderly, family members, and community members such as village health volunteers in health- promotion activities and skills training for the elderly in order to perform self-health care behaviors to prevent chronic disease were statistically significant. Such social support was also associated with the elderly's self-efficacy in terms of behavioral change, lifestyle, and satisfaction [13–15]. However, scant research on social participation to prevent hypertension in the elderly has

been conducted in which the study outcome has been measured concurrently with their caregivers. This research applied the concept of social support [16] as a conceptual framework for the organization of caregiver support activities in helping the elderly enhance their self-efficacy [17] to modify their prevention of hypertension behaviors, and to encourage participation among family members to create a better bond and joyful, caring moments in the family. The study's objective was to access a social support family caregiver training program aimed at improving health care behaviors, reducing high blood pressure and total cholesterol (TC), and increasing high- density lipoprotein (HDL) among older adults in the rural areas of Phayao province, Thailand.

## Methods

This study was a quasi-experimental pretest-posttest design with an intervention and control group based on the main project called "Unit of Excellence: Health Promotion and Quality of Life". This research was conducted from November 2020 to April 2021 in the rural areas of Phayao province, 30 kilometers from the city. The district does not have secondary hospitals to provide health services to the public. There are 15 sub-districts of Muang district, Phayao province. The researcher purposely selected two villages from two sub-districts using criteria on the rapidly growing number of elderly registered as members of a hypertensive patient risk group in the area. An intervention group and a control group were chosen at random from two villages to avoid any potential contamination. After that, simple random sampling was employed using the lottery method to determine the selection of participants for each group. However, the intervention and control groups are located 20 kilometers apart. With this relatively short distance, both groups shared geographical, social, and demographic similarities. It's plausible that this might have no or only a minor effect on the findings.

The criteria for selecting representative households from villages are as follows: 1) Publicized the research program through village health volunteers searching for volunteers to participate in the research, and 2) Pass the qualifications according to the criteria for both older adults and caregivers. The criteria for selecting elderly at risk of hypertension are as follows: 1) Aged over 60 years, male or female; 2) Diagnosed by a doctor as an at-risk hypertension patient with high systolic blood pressure (SBP) = 130–139 mmHg, and/or diastolic blood pressure (DBP) = 85–89 mmHg, or at risk level 1 with SBP = 140–159 mmHg and/or DBP = 90–99 mmHg [18] for more than one year; 3) No health complications or chronic non-communicable diseases and not on prescribed medication; 4) Having abnormal blood lipids and has not received medication also had a recent (past three months) lipid level test; 5) No mental or physical disorders such as dementia or paralysis; and 6) Voluntary participation in the research program. The criteria for selecting family caregivers were as follows: 1) Being a family member of an elderly person, such as a husband or wife, child, grandchild, or relative, 2) Being a primary caregiver of the elderly, 3) Living with the elderly in the same household for 3–4 hours per day, and 4) Willing to participate in the study.

The sample was determined by comparing two independent formula means [19]. The medium effect size was represented by 0.5, a power of 0.2 for the probability of rejecting a false null hypothesis, and alpha set at 0.05 (type I error). The minimum sample size for each group of this study was 64, increased by 5% to account for the expected dropout rate. As a result, the final sample size for each group was 67; and the total sample size was presented as 268 participants, as shown in Fig 1. A face-to-face interview with a questionnaire was used to collect data from the elderly and caregivers. Blood pressure readings and blood samples for cholesterol testing were taken from the elderly. The protocol was approved by the University of Phayao Human Ethics Committee, Thailand (No. 2/089/61). The trial was registered at Thai Clinical

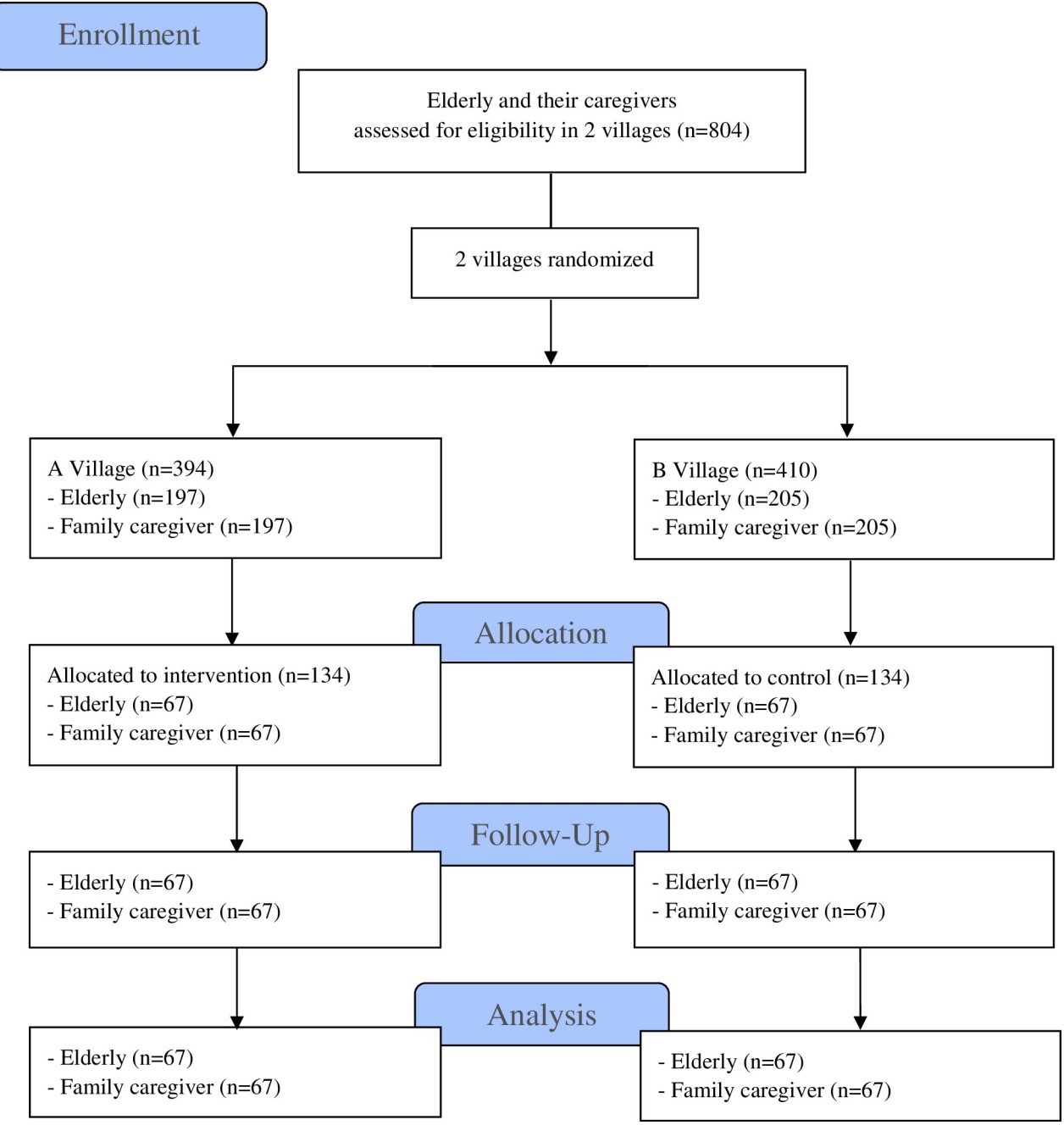

**Fig 1. CONSORT flow diagram of the study participants.**

Trials Registry (thaiclinicaltrials.org) and the registration number is TCTR20210203005. The trial was registered participants gave a written informed consent prior to data collection.

## Questionnaire

The questionnaires used to collect the quantitative data were applied from previous research and developed for appropriate use in the research study in the rural northern part of Thailand. The questionnaire for both the elderly and the caregivers comprised 4 four parts (see S1 File or

S2 File). Part 1) General demographic characteristics of the elderly included gender, age, marital status, education, income sufficiency, dietary habits/tasty food, types of physical activity, alcohol intake, smoking, body mass index (BMI), and health information. The questionnaire for caregivers included gender, age, marital status, education, occupation, income sufficiency, relationship with the elderly, alcohol intake, smoking, and information received. Part 2) Knowledge regarding hypertension based on previous studies [10], which consisted of 22 items. These could be classified into three areas; 1) Symptoms, severity, and complications; 2) Causes and factors; and 3) Prevention and control. The types of questions are choice-based with two answers: correct and wrong answers. The total score is in the range of 0–22 points. This part of the questionnaire was used by both the elderly and their caregivers. Part 3) Self-efficacy toward the prevention of hypertension among the elderly was modified to be appropriate for use in the context of the community [17, 20]. Similarly, healthcare self-efficacy among elderly patients with hypertension was also developed for caregivers. Both questionnaires consisted of 10 items with three rating scales: Agree, Not sure, and Disagree. The total score is in the range of 10–30 points. Part 4) Health care behavior toward hypertension among the elderly was modified from previous research [10, 18, 21–23]. For caregivers, toward healthcare for elderly patients with hypertension were adopted from previous studies [10]. These can be classified into three areas; 1) Control risk factors; 2) Prevention and control; and 3) Self-care behaviors. Both questionnaires consisted of 20 items with the following levels of rating scales: Never Practice, Practice Rarely (1–2 times/week), Practice Sometimes (3–4 times/week), and Practice Regularly (5–7 times/week). The total score is in the range of 20–80 points.

Three experts with experience of the aging population, non-communicable diseases, and public health examined the questionnaires for content validity. Then, the researchers tried out the questionnaire among 30 samples with identical characteristics. The reliability test of the part 2–4 questionnaire showed the Kuder-Richardson Formula (KR20) and Cronbach's alpha coefficients at between 0.79 and 0.83.

## Blood pressure (BP) and cholesterol measurement

An automatic blood pressure measurement device was used to measure blood pressure [18]. The device was attached to an airbag and cloth with an arm circumference of about 27–34 cm, to be wrapped around the arm [18]. Arterial blood pressure was measured according to the standardized procedure recommended by the World Health Organization [24]. After a rest of ten minutes, two sitting blood pressure readings were taken five minutes apart on either arm. The average of these two readings was used as the BP reading of the individual.

A blood lipid-level detector was used to detect the lipid level including total cholesterol and high-density lipoprotein (HDL-cholesterol)–among participants. A medical technician from the School of Allied Health Sciences, University of Phayao, performed blood collection, analysis, and interpretation. The laboratory equipment was inspected, and passed the quality inspection required.

## Social support family caregiver training program (SSFCTP)

The SSFCTP was initiated to implement social support and self-efficacy theory [16, 17], which aims to promote health among the elderly through family caregiver activities. The SSFCTP specifically encourages caregivers to promote self-efficacy and self-care among the elderly at risk of hypertension during research activities. The following people were recruited from the area to participate in the program: One public health scholar, one nurse, and one care manager team (four people), all of whom were to assist in research. The care manager team had to

undergo a 70-hour training course in elderly care held by the Ministry of Public Health [25]. The researcher educated the assistants for two hours on the use of research instruments for collecting data. The research assistants were subsequently tested on their knowledge of using the instruments for data collection. Notebooks and observations were used to ensure consistent understanding among the research team.

The intervention group was required to participate in the program once a week, for between 180 and 240 minutes, for a total of 12 weeks if it was convenient for the participants. This research program included two main activities: 1) Skills in building relationships between caregivers and the elderly, which included discussing problems encountered in everyday life; sharing past experiences related to caring for the elderly (30–40 minutes); and improving knowledge about hypertension through video media prepared by the researcher. The video's topics included pathology, mechanisms, causes, symptoms, and disease prevention. The program also focused on imparting knowledge related to health complications caused by or affecting the severity of the current health problem (120 minutes): 2) Both caregivers and elders received a diary and a self-care manual titled "Promise with Hearts" [16, 17] for recording their daily routine activities (BP levels, diet, exercise) designed by the researcher. The participants were also asked to conduct their own research to understand how (self-regulation) to prevent hypertension behaviors.

Activities for weeks 2 to 6 focused on self-efficacy skills, modeling, nutrition education, and physical activities. The activities are described in the following sub-activities:

a) Self-efficacy skills regarding prevention and control of hypertension. These involved building successful experiences, and having caregivers share their experiences in caring for the elderly, particularly regarding dieting, stress management, and exercise for the elderly. The session ran for 60 minutes.

b) Modeling was a technique employed to use during the session to enhance skills. The examples of modeling techniques included case studies, moderators describing successful experiences, and family caregivers and elderly involvement in supporting elderly care to prevent hypertension and health complications. The session ran for 120 minutes.

c) Nutrition education and health coaching for indoor home-based DASH (Dietary Approaches to Stop Hypertension) guidelines [20, 26] offer practical advice on gaining nutritional knowledge. This advice includes avoiding high-fat dairy products, sodium in food, understanding the health benefits of food, and choosing a dietary regimen that addresses health needs. DASH guidelines were applied for the elderly and caregivers so they could accumulate knowledge and become familiar with the concept of activities that enhance health. The activities included using small groups of caregivers and older adults practicing cooking together after selecting a local food menu. The session ran for 120 minutes.

d) Physical activity [27] included informative lectures on appropriate exercises for the elderly, having caregivers and the elderly join together to practice using music that is fun and enjoyable during exercises, and using the arms and legs to perform rhythmic moves or local retro-melody dances in small groups. These sessions ran for 90 minutes. Outside of class activities, elderly participants were encouraged to exercise at home through such activities as watering plants, cleaning the house, performing other house-related activities, or walking around the community.

Activities for weeks 7 to 12 focused on 1) Counseling skills; and 2) Follow up and reinforcement. Counseling skills emphasized communication between caregivers and the elderly,

especially communication relating to self-efficacy and self-empowerment among caregivers, exchanging and sharing information, listening skills, and consulting correctly. The session was done in small groups of caregivers and elderly participants, and ran for 120 minutes. Follow-up and reinforcement consisted of two home visits between 30 and 40 minutes. Telephone calls were used for participants who were unable to meet in person for 30 minutes.

Moreover, during the follow-up, caregivers were required to review the activities, assess their behaviors, and encourage self-efficacy to support care for the elderly, so the latter could practice behaviors that would prevent hypertension on their own. Participants in the control group received regular education from local health authorities where their health records were kept. Data collection was performed three times: at baseline, after 12 weeks (post-intervention), and three months after the end of the intervention (follow-up). A summary of the data collection is shown in Fig 2.

### Data analysis

Data were analyzed by SPSS Version 17, licensed from Chiang Mai University (SPSS Inc., Chicago, IL, USA). An independent t-test and chi-square test were used to find the differences in the general information of the elderly and their caregivers at baseline between the intervention and control groups for the continuous (age and BMI) and categorical variables, respectively. The effects of the SSFCTP were tested using repeated measures ANOVA. Our study inserted the command syntax with Bonferroni adjustment to test the difference between the intervention and control groups at each time juncture. All the statistical tests were given a p-value of 0.05 as the level of significance. Based on blood pressures as a primary outcome of this study, crosstabs were used to compute the relative risk (RR) of hypertension at each time point, defined as SBP $\geq$ 140 mmHg and/or DBP $\geq$ 90 mmHg. Dataset is presented in the S3 File.

### Results

There were 67 pairs of elderly people and their caregivers in each of the intervention and control groups. More than half of the elderly participants were females (62.7%), aged between 60 and 69 (66.4%). Elderly variables including age, gender, marital status, education level, income, dietary habits, types of physical activity, health information, alcohol intake, smoking, and BMI showed no statistically significant differences between the two groups at baseline (Table 1). When we analyzed the general demographic characteristics of caregivers from the intervention and control groups, we found that the majority of participants were in the age range 50–59 years (38.8%); and most of them were female (73.1%). In terms of the relationship to the patients, most family caregivers were offspring, including a child (30.6%) and a grandchild (30.6%). In both groups of caregivers, there were no statistically significant differences in age, gender, marital status, education level, working status, income, relationship to the patients, alcohol consumption, cigarette smoking, and health information (Table 2). These results showed that the elderly and caregivers in both groups shared similar characteristics.

The analysis of repeated measures ANOVA revealed the overall effects of the SSFCTP on changes in knowledge, self-efficacy, health care behaviors, and SBP and DBP among the elderly participants (p<0.05) (Table 3). The Greenhouse-Geisser correction for within-subject testing was used due to a violation of sphericity. An effect of the SSFCTP (depending on time) was found in knowledge, SBP, DBP, and total cholesterol level among the elderly (p<0.05) (Fig 3). In terms of caregiver outcomes, the results showed that the intervention program had a significant effect on changes in knowledge, self-efficacy, and behaviors toward health care for elderly patients with hypertension over three time points (p<0.05) (Fig 4).

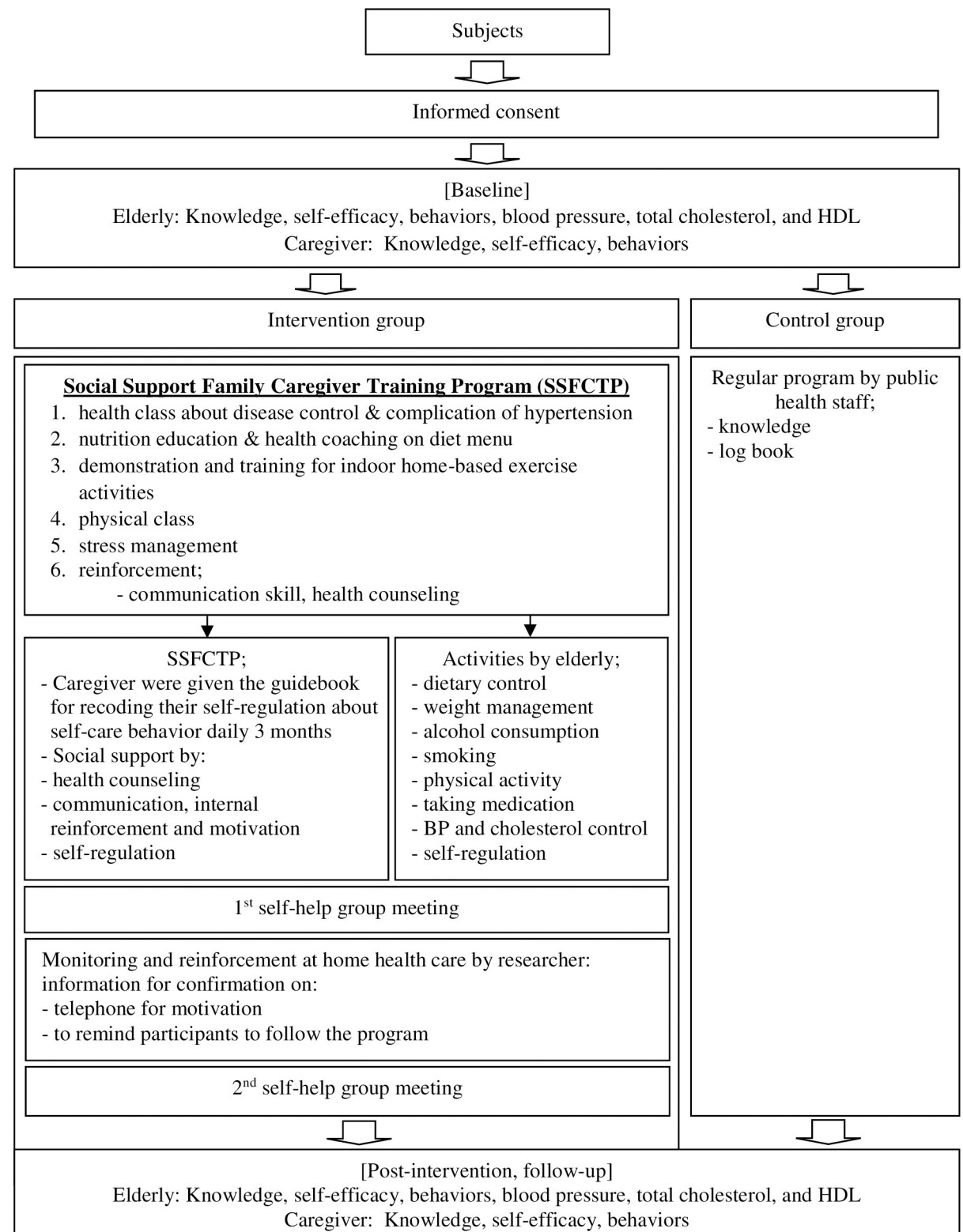

**Fig 2. Schedule of social support family caregiver training program.**

**Table 1. Number and percentage of elderly characteristics categorized between the intervention and control groups.**

| Elderly variable | Total (N = 134) | Intervention group (n = 67) | Control group (n = 67) | P-value |
|---|---|---|---|---|
| | n (%) | n (%) | n (%) | |
| Age | | | | 0.213 |
| Mean (SD) | 68.0 (7.5) | 68.8 (8.0) | 67.2 (7.1) | |
| Max—Min | 92–60 | 92–60 | 87–60 | |
| Gender | | | | 0.475 |
| Male | 50 (37.3) | 23 (34.3) | 27 (40.3) | |
| Female | 84 (62.7) | 44 (65.7) | 40 (59.7) | |
| Status | | | | 0.722[a] |
| Single | 4 (3.0) | 3 (4.5) | 1 (1.5) | |
| Marry | 85 (63.4) | 42 (62.7) | 43 (64.2) | |
| Separated/divorced/widow | 45 (33.6) | 22 (32.8) | 23 (34.3) | |
| Education level | | | | 0.119[a] |
| No education | 19 (14.2) | 6 (9.0) | 13 (19.4) | |
| Primary school | 56 (41.8) | 27 (40.2) | 29 (43.3) | |
| Secondary school | 57 (42.5) | 32 (47.8) | 25 (37.3) | |
| College/university | 2 (1.5) | 2 (3.0) | 0 (0.0) | |
| Income | | | | 0.489 |
| Insufficient | 70 (52.2) | 33 (49.3) | 37 (55.2) | |
| Sufficient | 64 (47.8) | 34 (50.7) | 30 (44.8) | |
| Dietary habit/tasty food | | | | 0.889 |
| No | 14 (10.4) | 6 (9.0) | 8 (11.9) | |
| Salt | 44 (32.8) | 23 (34.3) | 21 (31.3) | |
| Sugar | 54 (40.3) | 26 (38.8) | 28 (41.8) | |
| Fat | 22 (16.4) | 12 (17.9) | 10 (14.9) | |
| Types of physical activity | | | | 0.518 |
| Does not activity | 42 (31.3) | 20 (29.9) | 22 (32.8) | |
| Walk around the house every day | 51 (38.1) | 27 (40.3) | 24 (35.8) | |
| Housework (everyday) | 25 (18.7) | 10 (14.9) | 15 (22.4) | |
| Exercise (2–3 times per week) | 16 (11.9) | 10 (14.9) | 6 (9.0) | |
| Alcohol intake | | | | 0.325 |
| No | 99 (73.9) | 52 (77.6) | 47 (70.1) | |
| Yes | 35 (26.1) | 15 (22.4) | 20 (29.9) | |
| Smoking | | | | 0.547 |
| No | 101 (75.4) | 52 (77.6) | 49 (73.1) | |
| Yes | 33 (24.6) | 15 (22.4) | 18 (26.9) | |
| Body mass index (BMI) | | | | 0.768 |
| Mean (SD) | 23.3 (3.5) | 23.4 (3.5) | 23.2 (3.5) | |
| Max—Min | 30–17 | 30–17 | 30–17 | |
| Receiving information about HT | | | | 0.370 |
| No | 49 (36.6) | 22 (32.8) | 27 (40.3) | |
| Yes (public health officer, health volunteer, online media, family) | 85 (63.4) | 45 (67.2) | 40 (59.7) | |

[a] P-value are calculated using Exact test.

The effect of the SSFCTP within each level of time is presented in Table 4. When comparing the studied variables of the elderly and their caregivers before joining the intervention, we found there were no statistically significant differences in knowledge, self-efficacy, health care

**Table 2. Number and percentage of caregiver characteristics categorized between the intervention and control groups.**

| Caregiver variable | | Total (N = 134) | Intervention group (n = 67) | Control group (n = 67) | P-value |
|---|---|---|---|---|---|
| | | n (%) | n (%) | n (%) | |
| Age (years) | | | | | 0.604 |
| | Mean (SD) | 51.3 (8.0) | 51.7 (8.5) | 51 (7.5) | |
| | Max—Min | 68–32 | 68–32 | 63–34 | |
| Gender | | | | | 0.697 |
| | Male | 36 (26.9) | 19 (28.4) | 17 (25.4) | |
| | Female | 98 (73.1) | 48 (71.6) | 50 (74.6) | |
| Status | | | | | 0.602 |
| | Single | 15 (11.1) | 6 (9.0) | 9 (13.5) | |
| | Marry | 79 (59.0) | 42 (62.7) | 37 (55.2) | |
| | Separated/divorced/widow | 40 (29.9) | 19 (28.3) | 21 (31.3) | |
| Education level | | | | | 0.875[a] |
| | No education | 27 (20.1) | 12 (17.9) | 15 (22.4) | |
| | Primary school | 61 (45.5) | 30 (44.8) | 31 (46.3) | |
| | Secondary school | 37 (27.6) | 20 (29.9) | 17 (25.4) | |
| | College/university | 9 (6.8) | 5 (7.5) | 4 (6.0) | |
| Working status | | | | | 0.384 |
| | Not working | 59 (44.0) | 32 (47.8) | 27 (40.3) | |
| | Currently working | 75 (56.0) | 35 (52.2) | 40 (59.7) | |
| Income | | | | | 0.489 |
| | Insufficient | 70 (52.2) | 33 (49.3) | 37 (55.2) | |
| | Sufficient | 64 (47.8) | 34 (50.7) | 30 (44.8) | |
| Relationship to the patient | | | | | 0.090 |
| | Spouse | 28 (20.9) | 19 (28.4) | 9 (13.4) | |
| | Child | 41 (30.6) | 22 (32.8) | 19 (28.4) | |
| | Grandchild | 41 (30.6) | 17 (25.4) | 24 (35.8) | |
| | Others (other relative, friend, neighbor, health volunteer) | 24 (17.9) | 9 (13.4) | 15 (22.4) | |
| Alcohol drinking | | | | | 0.600 |
| | No | 77 (57.5) | 40 (59.7) | 37 (55.2) | |
| | Yes | 57 (42.5) | 27 (40.3) | 30 (44.8) | |
| Smoking | | | | | 0.701 |
| | No | 96 (71.6) | 47 (70.1) | 49 (73.1) | |
| | Yes | 38 (28.4) | 20 (29.9) | 18 (26.9) | |
| Receiving information about HT | | | | | 0.568 |
| | No | 39 (29.1) | 18 (26.9) | 21 (31.3) | |
| | Yes (public health officer, health volunteer, online media) | 95 (70.9) | 49 (73.1) | 46 (68.7) | |

[a] P-value are calculated using Exact test.

behaviors, SBP, DBP, total cholesterol, and HDL between the intervention and control groups. At the end of the intervention and follow-up period, the mean score of knowledge, self-efficacy, and health care behaviors in the intervention group was higher than that in the control group (p<0.05). In addition, SBP, DBP, and total cholesterol of elderly in the intervention group were statistically significantly lower compared to the control group (p<0.05). As measured by RR, the elderly who received SSFCTP were 1.114 times more likely to have hypertension than those who did not at baseline. Whereas RR was 0.730 times at post-intervention, and 0.313 times at follow-up (Fig 5).

**Table 3. Repeated measures ANOVA of the elderly and caregiver outcomes between the intervention and control groups.**

| Variable | | F-test | P-value | Partial Eta squared |
|---|---|---|---|---|
| Elderly' knowledge (score) | | | | |
| | Intervention | 11.80 | 0.001* | 0.082 |
| | Time[a] | 243.78 | <0.001* | 0.649 |
| | Intervention x Time[a] | 19.18 | <0.001* | 0.127 |
| Elderly' self-efficacy (score) | | | | |
| | Intervention | 5.98 | 0.016* | 0.043 |
| | Time[a] | 349.55 | <0.001* | 0.726 |
| | Intervention x Time[a] | 1.93 | 0.156 | 0.014 |
| Elderly' behaviors (score) | | | | |
| | Intervention | 9.49 | 0.003* | 0.067 |
| | Time[a] | 208.10 | <0.001* | 0.612 |
| | Intervention x Time[a] | 1.53 | 0.221 | 0.011 |
| SBP (mmHg) | | | | |
| | Intervention | 9.98 | 0.002* | 0.070 |
| | Time[a] | 427.55 | <0.001* | 0.764 |
| | Intervention x Time[a] | 71.53 | <0.001* | 0.351 |
| DBP (mmHg) | | | | |
| | Intervention | 11.79 | 0.001* | 0.082 |
| | Time[a] | 296.51 | <0.001* | 0.692 |
| | Intervention x Time[a] | 21.95 | <0.001* | 0.143 |
| Total cholesterol (mg/dL) | | | | |
| | Intervention | 1.46 | 0.230 | 0.011 |
| | Time[a] | 184.80 | <0.001* | 0.583 |
| | Intervention x Time[a] | 18.08 | <0.001* | 0.120 |
| HDL (mg/dL) | | | | |
| | Intervention | 0.45 | 0.502 | 0.003 |
| | Time[a] | 167.78 | <0.001* | 0.560 |
| | Intervention x Time[a] | 2.39 | 0.113 | 0.018 |
| Caregivers' knowledge (score) | | | | |
| | Intervention | 5.28 | 0.023* | 0.038 |
| | Time[a] | 271.59 | <0.001* | 0.673 |
| | Intervention x Time[a] | 13.34 | <0.001* | 0.092 |
| Caregivers' self-efficacy (score) | | | | |
| | Intervention | 7.86 | 0.006* | 0.056 |
| | Time[a] | 609.20 | <0.001* | 0.822 |
| | Intervention x Time[a] | 10.88 | <0.001* | 0.076 |
| Caregivers' behaviors (score) | | | | |
| | Intervention | 5.80 | 0.017* | 0.042 |
| | Time[a] | 447.57 | <0.001* | 0.772 |
| | Intervention x Time[a] | 14.70 | <0.001* | 0.100 |

[a] Greenhouse-Geisser correction.

* Significance at 0.05 level.

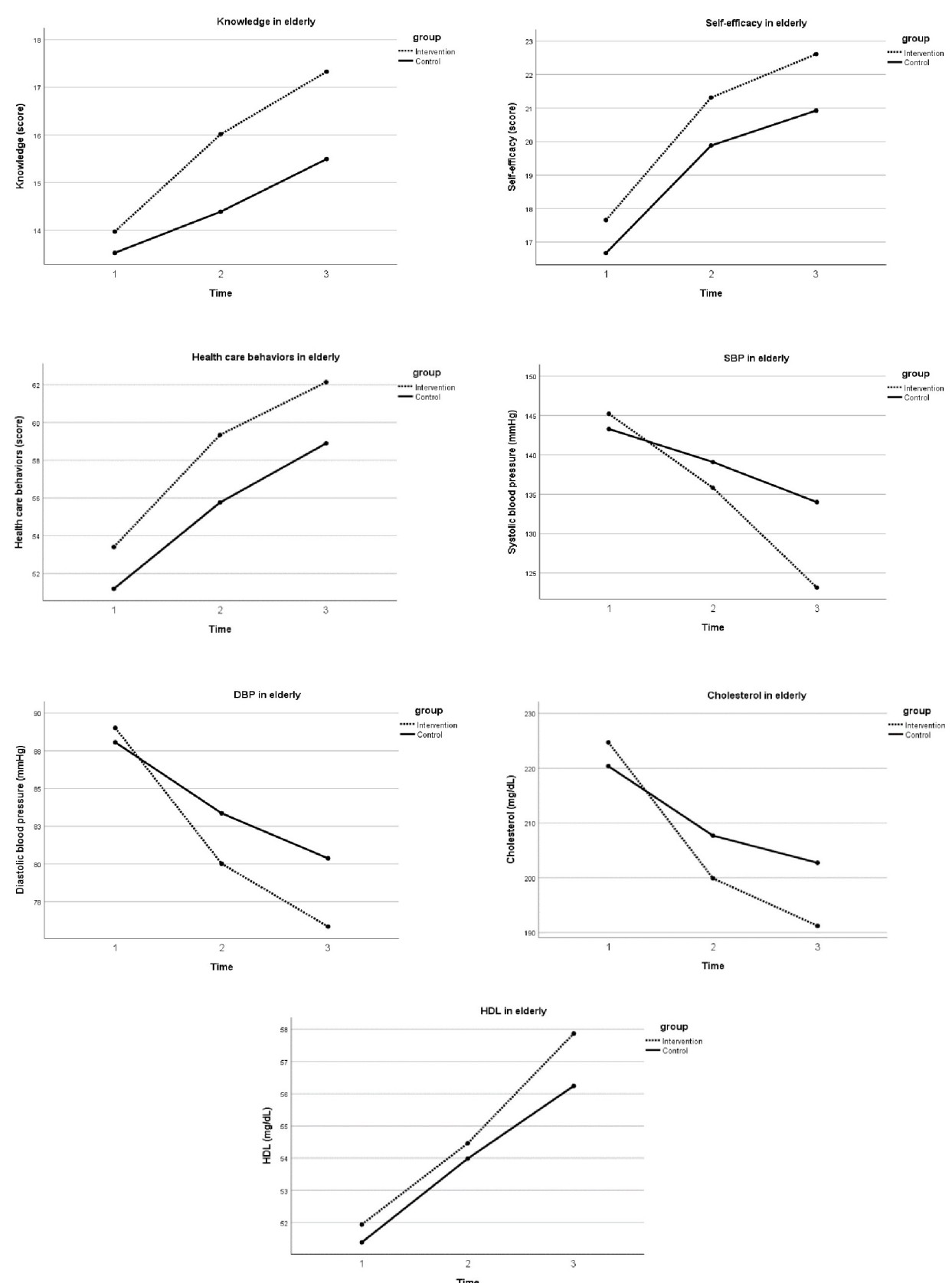

**Fig 3. Comparison of variables among elderly between two groups.**

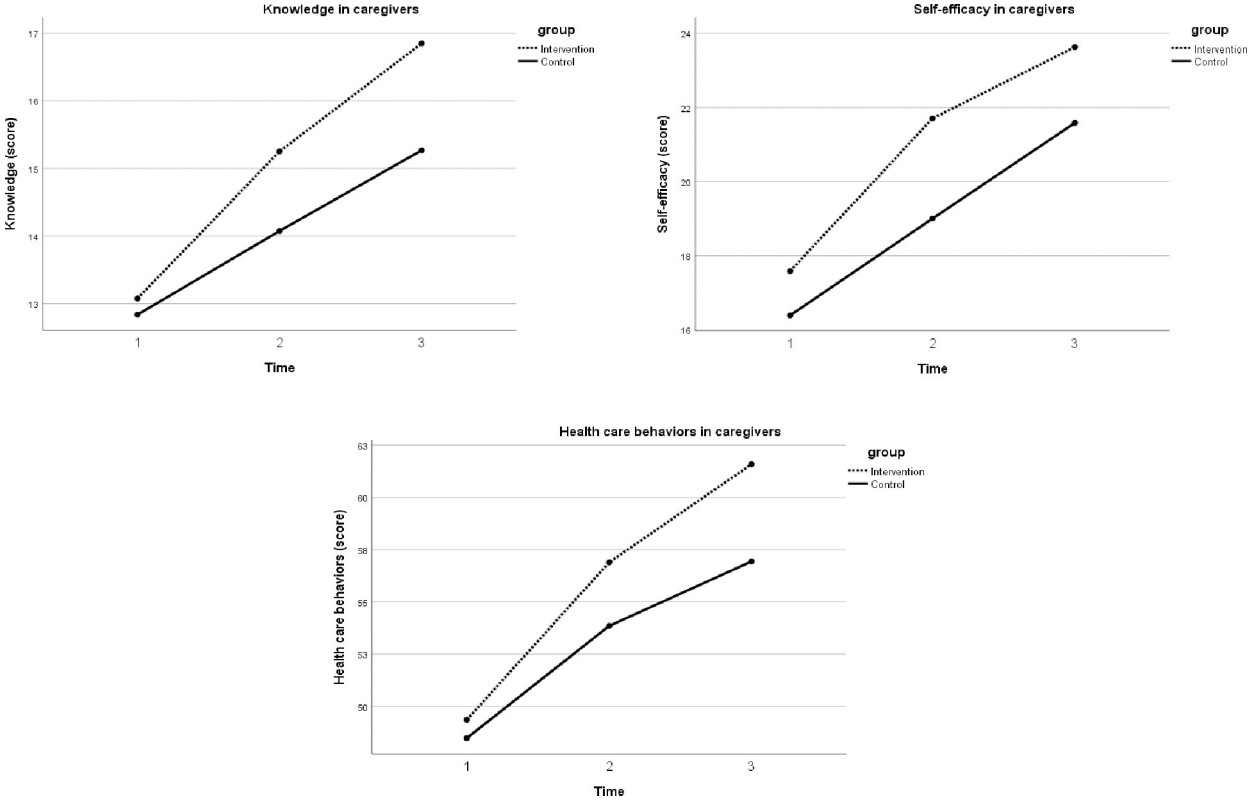

**Fig 4. Comparison of variables among family caregivers between two groups.**

## Discussion

The findings show that the "Social Support Family Caregiver Training Program (SSFCTP)" can help the elderly in rural areas control blood pressure, reduce total cholesterol, and improve knowledge, self-efficacy, and health care behaviors toward hypertension compared to the control group that received the regular program. At the same time, the SSFCTP can improve knowledge, self-efficacy, and healthcare behaviors for elderly patients with hypertension among their family caregivers. Only high-density lipoproteins (HDL) of the elderly were not statistically significantly different between the intervention and control groups at the end of the intervention and follow-up. However, the elderly participants had a higher level of HDL, after joining the SSFCTP.

Elders' knowledge, self-efficacy, and behaviors improved after participating in the SSFCTP and continued to work with their caregivers. One reason for this improvement could be attributed to videos used in the program that emphasized the severity of the diseases and health complications. In addition, the program focused on empowering caregivers to communicate, motivate, raise awareness, and encourage the older adults to control their blood pressure and monitor their health by recording their daily activities in a diary. These activities helped the elderly participants keep track of daily routine activities that they could share with the caregivers. This is consistent with the social support concept, which explains that being encouraged by families or healthcare workers to provide information, knowledge, advice, and reinforcement will enhance a patient's behaviors [16]. Like self-efficacy theory, if a person has a high perception or belief in their abilities, they will achieve the expected outcomes [17]. This is similar to a study that found that knowledge among participants joining an intervention program

**Table 4. Comparison of the elderly and caregiver outcomes between the intervention and control groups at baseline, post-intervention, and follow-up.**

| Variable | | All (n = 134) | Intervention group (n = 67) | Control group (n = 67) | Mean difference (SE) | P-value |
|---|---|---|---|---|---|---|
| | | Mean (SD) | Mean (SD) | Mean (SD) | | |
| Elderly' knowledge (score) | | | | | | |
| | Baseline | 13.7 (2.2) | 14 (2.3) | 13.5 (2.1) | 0.5 (0.4) | 0.239 |
| | Post-intervention | 15.2 (2.4) | 16.0 (2.5) | 14.4 (2.1) | 1.6 (0.4) | <0.001* |
| | Follow-up | 16.4 (2.7) | 17.3 (2.3) | 15.5 (2.7) | 1.8 (0.4) | <0.001* |
| Elderly' self-efficacy (score) | | | | | | |
| | Baseline | 17.2 (3.7) | 17.7 (3.6) | 16.7 (3.8) | 1.0 (0.6) | 0.125 |
| | Post-intervention | 20.6 (3.5) | 21.3 (3.5) | 19.9 (3.4) | 1.4 (0.6) | 0.017* |
| | Follow-up | 21.8 (3.3) | 22.6 (3.0) | 20.9 (3.4) | 1.7 (0.6) | 0.003* |
| Elderly' behaviors (score) | | | | | | |
| | Baseline | 52.3 (7.2) | 53.4 (7.0) | 51.2 (7.2) | 2.2 (1.2) | 0.074 |
| | Post-intervention | 57.6 (6.4) | 59.3 (6.3) | 55.8 (6.0) | 3.6 (1.1) | <0.001* |
| | Follow-up | 60.5 (5.7) | 62.1 (5.1) | 58.9 (5.8) | 3.2 (0.9) | <0.001* |
| SBP (mmHg) | | | | | | |
| | Baseline | 144.2 (8.2) | 145.2 (8.4) | 143.3 (7.9) | 1.9 (1.4) | 0.171 |
| | Post-intervention | 137.5 (7.9) | 135.8 (8.3) | 139.1 (7.1) | -3.3 (1.3) | 0.016* |
| | Follow-up | 128.6 (10.4) | 123.1 (10.0) | 134.0 (7.5) | -10.9 (1.5) | <0.001* |
| DBP (mmHg) | | | | | | |
| | Baseline | 88.5 (3.8) | 89.0 (3.9) | 88.1 (3.6) | 0.9 (0.7) | 0.147 |
| | Post-intervention | 81.7 (5.7) | 80.0 (5.0) | 83.4 (5.9) | -3.3 (0.9) | 0.001* |
| | Follow-up | 78.1 (5.6) | 75.8 (5.5) | 80.4 (4.7) | -4.5 (0.9) | <0.001* |
| Total cholesterol (mg/dL) | | | | | | |
| | Baseline | 222.5 (34.5) | 224.7 (39.1) | 220.4 (29.4) | 4.3 (6.0) | 0.470 |
| | Post-intervention | 203.8 (21.4) | 199.9 (23.1) | 207.7 (18.9) | -7.8 (3.6) | 0.034* |
| | Follow-up | 197.0 (19.1) | 191.2 (20.1) | 202.7 (16.2) | -11.5 (3.2) | <0.001* |
| HDL (mg/dL) | | | | | | |
| | Baseline | 51.7 (8.9) | 51.9 (7.9) | 51.4 (9.8) | 0.5 (1.5) | 0.721 |
| | Post-intervention | 54.2 (7.7) | 54.5 (6.6) | 54.0 (8.7) | 0.5 (1.3) | 0.722 |
| | Follow-up | 57.0 (6.7) | 57.9 (5.5) | 56.2 (7.7) | 1.6 (1.2) | 0.164 |
| Caregivers' knowledge (score) | | | | | | |
| | Baseline | 13.0 (3.0) | 13.1 (3.1) | 12.8 (3.0) | 0.2 (0.5) | 0.650 |
| | Post-intervention | 14.7 (2.8) | 15.3 (2.7) | 14.1 (2.9) | 1.2 (0.5) | 0.016* |
| | Follow-up | 16.1 (2.2) | 16.9 (2.1) | 15.3 (2.1) | 1.6 (0.4) | <0.001* |
| Caregivers' self-efficacy (score) | | | | | | |
| | Baseline | 17.0 (4.4) | 17.6 (4.4) | 16.4 (4.2) | 1.2 (0.7) | 0.113 |
| | Post-intervention | 20.3 (4.7) | 21.7 (5.0) | 19.0 (4.0) | 2.7 (0.8) | 0.001* |
| | Follow-up | 22.6 (4.0) | 23.6 (4.1) | 21.6 (3.5) | 2.0 (0.7) | 0.003* |
| Caregivers' behaviors (score) | | | | | | |
| | Baseline | 48.9 (7.7) | 49.4 (8.1) | 48.5 (7.4) | 0.9 (1.3) | 0.512 |
| | Post-intervention | 55.4 (7.9) | 56.9 (8.3) | 53.9 (7.2) | 3.0 (1.3) | 0.026* |
| | Follow-up | 59.3 (6.5) | 61.6 (6.3) | 56.9 (5.9) | 4.7 (1.1) | <0.001* |

* Significance at 0.05 level.

was significantly higher after the program [28]. A previous study found that after joining an intervention program that "helps family members", participant's demonstrated increased self-care confidence. Furthermore, they increased their exercise behavior, and lowered their blood pressure levels [14]. Caregivers in the current study are considered to be family members or

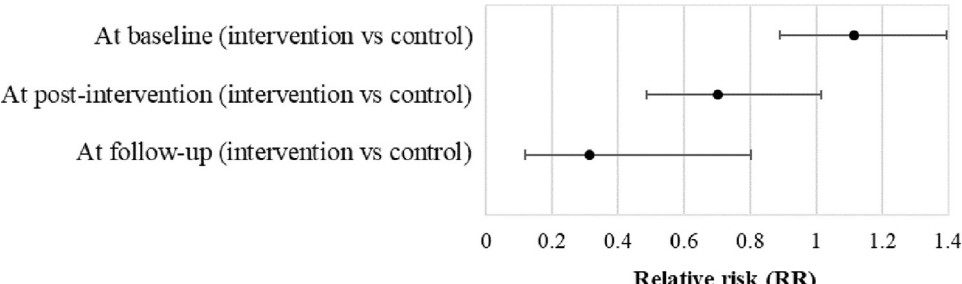

**Fig 5. Relative risk of hypertension in elderly receiving SSFCTP compared to controls.**

those who are closest to the patient, and their ages range between 50–69 years old [13]. In most cases, the family caregivers are primarily spouses or offspring. They know the behavior of the elderly patients, and their role is to contribute to the education of the elderly to modify their behaviors to prevent hypertension and health complications accordingly [10, 29]. Another study has shown the importance of family caregivers as contributors to education as in their capacity as intermediaries transferring knowledge between themselves and the patients [30].

In this study, the mean scores of health variables related to the prevention of hypertension increased, explaining that the cognitive performance of the elderly was associated with self-efficacy and knowledge gained from the program. Both the elderly and caregivers participated in small group activities to focus on their self-efficacy in preventing complications arising from hypertension. Prevention measures included, reducing their intake of salty foods, cutting down on high-fat foods, avoiding cigarettes and alcohol, managing stress, and exercising more regularly. The study showed that after participating in the activities, the change among elderly and their caregivers was consistent with one study that mentioned that a person who wishes to be successful must have a successful experience, according to the concept of verbal persuasion [17]. The implementation of the Social Support Family Caregiver Training Program intro-duced a "Promise of Hearts" activity to show concern for group members and allow caregivers and elderly people to ask questions, and to help them realize that they can be successful if they practice [17, 31]. The "Promise of Hearts" activity was designed to show concern for members of the group, in which verbal persuasion is effective when a successful action is linked to the subject. Our study found, that after participating in a self-care promotion program, a model activity and family involvement in meal planning resulting in the overall self-efficacy score was statistically higher than before the trial [32]. Moreover, another study, found that stimulation and reinforcement among family caregivers can result in hypertensive patients increasing their expectations of self-efficacy and motivate them to continue exercising regularly [14, 20].

The results revealed an increase in the mean scores of health care behaviors toward hyper-tension. It can be seen that the SSFCTP contributed to the prevention of hypertension, as the activities focused on the caregivers and the elderly. The activities also helped caregivers moti-vate dietary changes among the elderly. The DASH menu focused on local food according to the principle of "reduce salty, reduce disease" and increased exercise, which was evident in the intervention group. Participants in the intervention group exercised at least three times a week which was more than the times the control group exercised. The change in behavior is based on the concept that an individual's expectations for success must be motivated and reinforced by family members [16]. This resonates with a previous study, which found that a family care-giver takes care of a patient an average of 22 hours a week and is responsible for the patient's care over a long time [29, 33]. Caregivers should be knowledgeable and trained, and access health resources and innovative care for patients [33]. This is consistent with a study which

revealed that caregivers who received training were found to experience a statistically significant difference in knowledge and skills, and were able, to effect higher mean scores of self-care behavior among the elderly [14, 34]. Knowledge relating to the prevention of hypertension is essential for family caregivers. Having correct knowledge will assist caregivers in providing appropriate care and monitoring health risks and complications that may arise in the elderly [10, 34].

Lower blood pressure among the elderly illustrated that during self-directed activities, participants applied self-regulation by documenting local food menus, salt ratio, sodium, a DASH diet, and daily exercise. At the same time, the involvement of caregivers encouraging and reinforcing such self-regulation led to better control of blood pressure. These self-regulated activities should be encouraged and continued for self-care behavior. Similar to the study, family support for primary caregivers by encouraging, stimulating, and reinforcing joint activities led to increased exercise behavior among patients [14]. In addition, the follow-up period included home and telephone visits by the caregivers to provide encouragement and positive reinforcement to the elderly, which is consistent with the idea that personal enhancement will occur and must consist of successful action on its own. Observations from a model or other people's experiences and emotional stimulation can foster confidence in the practice of self-care behavior [17]. Previous research has found that diet and exercise can lead to lower blood pressure levels. This is because exercise aids the secretion of nitric acid oxide in the body, which dilates blood vessels lowering resistance in the blood vessels causes blood pressure levels to drop accordingly [35]. Many previous studies have found that activities focusing on exchanging experiences include dietary models, dietary journaling, and home visitation to stimulate the effect on systolic blood pressure levels in the program groups [33, 36].

In terms of total cholesterol level, there was a difference among the three tests. The changes could be explained by the fact that the SSFCTP focused on getting experimental groups to select regional local foods based on the DASH menu. The caregivers could cook such foods for the elderly daily and empower the elderly by exchanging information and making agreements with them regarding self-care. In addition, caregivers and elders joined together to practice music exercises that were fun and joyful.

During the exercises, elders danced or moved to the rhythm of music played for them. The retro-melody dance can be carried out at home for at least 30 minutes a day. The caregivers also assessed the daily activities of the elderly. Previous studies, have found that a traditional combination exercise with moderate intensity pulsation, combined with at least 30–45 minutes of physical and mental exercise, results in a decrease in blood pressure and the percentage lipid profile [27]. Self-regulation by taking physical activity notes or doing physical activities at home results in lower body fat levels; moreover, regular physical exercise reduces body fat [27, 37, 38]. Although the hypothesis was not followed when the HDL test was performed, HDL levels were higher and better than the control group during the follow-up period. The SSFCTP may contribute to helping optimally maintain and improves HDL levels. Also, this suggests that the results should be observed for a longer time. The process should take time because it HDL levels increase or decrease depending on past personal behaviors and factors that may affect HDL levels [27]. This is consistent with previous studies, which have found that family caregiver groups had higher HDL levels than the control groups [39].

## Limitations

First, this research study was conducted during the COVID-19 pandemic, during which; caregivers may have had other community activities to participate in. For this reason, they may not have been able to record routine activities in their diaries regularly. In this case, caregivers had

to be reminded by the research assistant to keep recording routinely. Secondly, this study was conducted among older adults at risk of hypertension. Most could perform daily activities typically, which could be a confounding factor affecting the research results. The researcher controlled for such factors by designing two groups to test the variables for the accuracy of the results. Lastly, as the SSFCTP was performed in rural areas of Phayao Province, the study results may not represent other populations. However, they can be applied to individuals at risk of hypertension in rural areas with similar physical characteristics; and they can be applied in conjunction with social support ideas. One of the strengths of this study is the participation of family members in helping lower the blood pressure of groups of elderly at risk. This participation is the primary source of social support for the elderly because caregivers are people in the family who have close relationships with the monitored, and can closely monitor those at risk of hypertension. Secondly, education in small groups allows the elderly and their caregivers to share problems and experiences, communicate more about health, interests, and enthusiasm in answering questions. Moreover, they can have more exercise time, get involved together, and interact with each other. In addition, neighbors can help the elderly increase their ordinary daily activities; thus, the elderly may experience joyful and fun exercise activities with people in the neighborhood. A further study may be required to investigate additional clinical outcomes such as low-density lipoprotein, triglycerides, and physical examination, including waist measurement, to look at long-term changes and their association with complications of other diseases. Therefore, the program activities could be extended by at least 9–12 months.

## Conclusions

This study described the effect of the social support family caregiver training program (SSFCTP) on controlling blood pressure, reducing total cholesterol, and improving self-care behaviors among older adults in rural areas at risk of hypertension. The results also showed that the participation among the elderly and caregivers in the activities program could increase knowledge and, self-efficacy in personal care of the elderly in the family, and provide confidence to strengthen the relationship and communicate with family members on self-care. This study highlights the importance of caregivers and the long-term benefits for relevant organizations at the community level, including health-promoting hospitals and local administrative organizations to guide and support the health promotion activities for family members and patients with chronic health conditions in the community.

## Supporting information

**S1 File. Questionnaire in the Thai language.**
(DOCX)

**S2 File. Questionnaire in the English language.**
(DOCX)

**S3 File. Dataset.**
(XLSX)

## Acknowledgments

The author's gratitude and appreciation goes to administration of the Health Promoting Hospitals Sub-District, Maeung district, Phayao Province, community elderly, caregiver, and data collectors.

## Author Contributions

**Conceptualization:** Sorawit Boonyathee, Katekaew Seangpraw.

**Data curation:** Prakasit Tonchoy, Supakan Kantow.

**Formal analysis:** Parichat Ong-Artborirak, Pitakpong Panta, Dech Dokpuang.

**Funding acquisition:** Katekaew Seangpraw.

**Methodology:** Nisarat Auttama, Sasivimol Bootsikeaw, Monchanok Choowanthanapakorn.

**Project administration:** Katekaew Seangpraw.

**Resources:** Parichat Ong-Artborirak, Pitakpong Panta.

**Software:** Sorawit Boonyathee, Sasivimol Bootsikeaw.

**Visualization:** Nisarat Auttama, Monchanok Choowanthanapakorn.

**Writing – original draft:** Katekaew Seangpraw, Parichat Ong-Artborirak.

**Writing – review & editing:** Sorawit Boonyathee, Katekaew Seangpraw, Parichat Ong-Artbor-irak, Nisarat Auttama, Prakasit Tonchoy, Supakan Kantow, Sasivimol Bootsikeaw, Mon-chanok Choowanthanapakorn, Pitakpong Panta, Dech Dokpuang.

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
