## [Decision Letter · Decision Letter 0]

31 Aug 2021

PONE-D-21-22371

Effects of a Social Support Family Caregiver Training Program on Changing Blood Pressure and Lipid Levels among Elderly at risk of Hypertension in a Northern Thai Community

PLOS ONE

Dear Dr. Seangpraw,

Thank you for submitting your manuscript to PLOS ONE. After careful consideration, we feel that it has merit but does not fully meet PLOS ONE’s publication criteria as it currently stands. Therefore, we invite you to submit a revised version of the manuscript that addresses the points raised during the review process.

We look forward to receiving your revised manuscript.

Kind regards,

Yoshihiro Fukumoto

Academic Editor

PLOS ONE

Journal Requirements:

2. Thank you for submitting your clinical trial to PLOS ONE and for providing the name of the registry and the registration number. The information in the registry entry suggests that your trial was registered after patient recruitment began. PLOS ONE strongly encourages authors to register all trials before recruiting the first participant in a study.

1) your reasons for your delay in registering this study (after enrolment of participants started);

2) confirmation that all related trials are registered by stating: “The authors confirm that all ongoing and related trials for this drug/intervention are registered”.

“The authors are grateful to the research project was supported by the Thailand science research and innovation fund and the University of Phayao the Unit of Excellence named “Health Promotion and Quality of Life” grant number FF64-UoE009.”

“The authors are grateful to the research project was supported by the Thailand science research and innovation fund and the University of Phayao the Unit of Excellence named “Health Promotion and Quality of Life” grant number FF64-UoE009.”

We note that you have provided funding information within the Acknowledgements Section. Please note that funding information should not appear in the Acknowledgments section or other areas of your manuscript. We will only publish funding information present in the Funding Statement section of the online submission form.

“The authors are grateful to the research project was supported by the Thailand science research and innovation fund and the University of Phayao the Unit of Excellence named “Health Promotion and Quality of Life” grant number FF64-UoE009.”

6. We note that you have indicated that data from this study are available upon request. PLOS only allows data to be available upon request if there are legal or ethical restrictions on sharing data publicly. For information on unacceptable data access restrictions, please see http://journals.plos.org/plosone/s/data-availability#loc-unacceptable-data-access-restrictions.

Reviewers' comments:

Reviewer's Responses to Questions

**Comments to the Author**

1. Is the manuscript technically sound, and do the data support the conclusions?

Reviewer #1: Partly

Reviewer #2: Yes

Reviewer #3: Partly

2. Has the statistical analysis been performed appropriately and rigorously? 

Reviewer #1: Yes

Reviewer #2: Yes

Reviewer #3: No

3. Have the authors made all data underlying the findings in their manuscript fully available?

Reviewer #1: Yes

Reviewer #2: Yes

Reviewer #3: Yes

4. Is the manuscript presented in an intelligible fashion and written in standard English?

Reviewer #1: Yes

Reviewer #2: Yes

Reviewer #3: Yes

5. Review Comments to the Author

Reviewer #1: This randomized interventional study evaluates the effects of a Social Support Family Caregiver Training Program (SSFCTP) on changing blood pressure and lipid levels among elderly at risk of hypertension. This is an interesting study.

My major concern is that in intervention arm, the authors conducted not only SSFCTP but also weight management, alcohol consumption, smoking, physical activity, taking medication, BP and cholesterol control, and self-regulation. How the authors purify the SSECTP effect in this setting?

Reviewer #2: This paper was a clinical trial study in a northern Thai community. This study was showed that elderly people at risk of hypertension increased knowledge about hypertension using SSFCTP and importance of caregivers.

Although this manuscript is interesting and meaningful, there are some concerns in it.

Major comments

1) The most serious concern is a short-term study in 12 weeks. We would like to know odds ratio or hazard ratio for morbidity of hypertension. Increasing knowledge about hypertension keeps blood pressure good, and prevent of hypertension will be make it decreased. This paper should focus on blood pressure.

2) Are there any clinical implications in your study? I think SSFCTP is a meaningful program. But SSFCTP has several contents such as knowledge, efficacy, and behavior. Which content was a most important for health and blood pressure?

3) Table4 and Table5 were so complicated. These tables should be represented in figures as you emphasized variables.

Minor comments

In line at 477, the paper was written TSSFCP. Does it mean SSFCTP?

Reviewer #3: Present study corresponds to two arm parallel group randomized controlled trial. I have several reservations about the study which needs further explanations.

1. It looks like two groups (treatment and control) are collected from two entirely different village/region. This mean the effect due to particular location cannot be eliminated. This is true randomization authors must explain why this is done and the possible weakness of the findings.

2. Elderly and caregiver are they belong to the same family, or they are chosen at random is not clear. If they are form same family they from a dyad which must be accounted for during statistical modelling.

3. The most serious concern is the power analysis presented in the paper (line 134-140). It states a two-sample t-test is used to power the study. Yet the main analysis proposed is RMANOVA. Repeated measure ANOVA is the correct model for longitudinal design, however, same model must be used for power analysis, which is not done. The paper also report pre-post type design for which t-test can be used for power analysis, but then RMANOVA should not be used for data analysis. More importantly region effect and dyad effect (if true) should be part of the model building. From the table 3 reporting it is clear RMANOVA is the right model but the study is not powered properly.

4. What about missing data and multiple testing issues, which is not addressed sufficiently. Please clearly stated which one is/are primary outcome/s. In table 3 multiple interaction term is tested, which means multiple testing correction must be used as otherwise type-1 error rate will be inflated. This may indicate the study is under-powered.

6. PLOS authors have the option to publish the peer review history of their article (what does this mean?). If published, this will include your full peer review and any attached files.

Reviewer #1: No

Reviewer #2: No

Reviewer #3: No

---

## [Author Response · Author response to Decision Letter 0]

25 Sep 2021

Dear Reviewers,

According to submit the manuscript title ““Effect of a Social Support Family Caregiver Training Program on Changing Blood Pressure and Lipid Levels Among Elderly with Hypertension Risk in Northern Thai Community” PONE-D-21-22371 - [EMID:66137e20164778db].”. We’ve improved according to the reviewer's recommendations as;

Reviewer 1: I have incorporated all of your suggestions into mu revision. They have been very helpful and important. Thank you.

Reviewer 2: I have incorporated all of your suggestions into mu revision. They have been very helpful and important. Thank you.

Reviewer : I have incorporated all of your suggestions into mu revision. They have been very helpful and important. Thank you.

If you need further information, please do not hesitate to contact me.

Thank you for kindness.

Best Regards,

Katekaew

---

## [Decision Letter · Decision Letter 1]

25 Oct 2021

Effects of a Social Support Family Caregiver Training Program on Changing Blood Pressure and Lipid Levels among Elderly at risk of Hypertension in a Northern Thai Community

PONE-D-21-22371R1

Dear Dr. Seangpraw,

We’re pleased to inform you that your manuscript has been judged scientifically suitable for publication and will be formally accepted for publication once it meets all outstanding technical requirements.

Kind regards,

Yoshihiro Fukumoto

Academic Editor

PLOS ONE

Additional Editor Comments (optional):

Reviewers' comments:

Reviewer's Responses to Questions

**Comments to the Author**

1. If the authors have adequately addressed your comments raised in a previous round of review and you feel that this manuscript is now acceptable for publication, you may indicate that here to bypass the “Comments to the Author” section, enter your conflict of interest statement in the “Confidential to Editor” section, and submit your "Accept" recommendation.

Reviewer #1: All comments have been addressed

Reviewer #2: All comments have been addressed

2. Is the manuscript technically sound, and do the data support the conclusions?

Reviewer #1: (No Response)

Reviewer #2: Yes

3. Has the statistical analysis been performed appropriately and rigorously? 

Reviewer #1: (No Response)

Reviewer #2: Yes

4. Have the authors made all data underlying the findings in their manuscript fully available?

Reviewer #1: (No Response)

Reviewer #2: Yes

5. Is the manuscript presented in an intelligible fashion and written in standard English?

Reviewer #1: (No Response)

Reviewer #2: Yes

6. Review Comments to the Author

Reviewer #1: (No Response)

Reviewer #2: (No Response)

7. PLOS authors have the option to publish the peer review history of their article (what does this mean?). If published, this will include your full peer review and any attached files.

Reviewer #1: No

Reviewer #2: No

---

## [Editor Report · Acceptance letter]

15 Nov 2021

PONE-D-21-22371R1 

Effects of a Social Support Family Caregiver Training Program on Changing Blood Pressure and Lipid Levels among Elderly at risk of Hypertension in a Northern Thai Community 

Dear Dr. Seangpraw:

I'm pleased to inform you that your manuscript has been deemed suitable for publication in PLOS ONE. Congratulations! Your manuscript is now with our production department. 

Kind regards, 

on behalf of

Dr. Yoshihiro Fukumoto 

Academic Editor

PLOS ONE